# Potential Application of High Hydrostatic Pressure on the Production of Hydrolyzed Proteins with Antioxidant and Antihypertensive Properties and Low Allergenicity: A Review

**DOI:** 10.3390/foods12030630

**Published:** 2023-02-02

**Authors:** Ana Paula Miguel Landim, Julia Hauck Tiburski, Caroline Grassi Mellinger, Pablo Juliano, Amauri Rosenthal

**Affiliations:** 1Embrapa Agroindústria de Alimentos, Rio de Janeiro 23020-470, RJ, Brazil; 2Postgraduate Program in Food Science and Technology, Federal Rural University of Rio de Janeiro (UFRRJ), Seropédica 23897-000, RJ, Brazil; 3Department of Food Technology, Federal Rural University of Rio de Janeiro (UFRRJ), Seropédica 23897-000, RJ, Brazil; 4CSIRO Agriculture and Food, Werribee, VIC 3030, Australia

**Keywords:** high hydrostatic pressure, enzymatic hydrolysis, antihypertensive activity, antioxidant capacity, allergenicity

## Abstract

The high hydrostatic pressure (HHP) process has been studied for several applications in food technology and has been commercially implemented in several countries, mainly for non-thermal pasteurization and shelf-life extension of food products. HHP processing has been demonstrated to accelerate proteolytic hydrolysis at a specific combination of pressure and pressure-holding time for a given protein source and enzyme. The enzymatic hydrolysis of proteins is a well-known alternative to producing biologically active peptides, with antioxidant and antihypertensive capacity, from different food protein sources. However, some of these protein sources contain allergenic epitopes which are often not degraded by traditional hydrolysis. Moreover, the peptide profile and related biological activity of a hydrolysate depend on the protein source, the enzymes used, the parameters of the proteolysis process (pH, temperature, time of hydrolysis), and the use of other technologies such as HHP. The present review aims to provide an update on the use of HHP for improving enzymatic hydrolysis, with a particular focus on studies which evaluated hydrolysate antihypertensive and antioxidant capacity, as well as residual allergenicity. Overall, HHP has been shown to improve the biological properties of hydrolysates. While protein allergenicity can be reduced with traditional hydrolysis, HHP can further reduce the allergenicity. Compared with traditional hydrolysis methods, HHP-assisted protein hydrolysis offers a greater opportunity to add value to protein-rich products through conversion into high-end hydrolysate products with enhanced nutritional and functional properties.

## 1. Introduction

Consumers are increasingly looking for foods containing substances that assist in health promotion. In this scenario, hydrolyzed proteins rich in biologically active peptides offer a pathway to meet the growing market demand for novel functional products and health promoting foods [1]. In some countries, peptides such as isoleucine-proline-proline (IPP) and valine-proline-proline (VPP) are approved for use in different commercial products such as sour milk products (Calpis^®^, Calpis Food Industry Co., Ltd., Tokyo, Japan) and margarine (Becel Pro-Activ^®^, Evolus Co., Ltd., Bern, Switzerland) [2]. In this context, the development of technologically and economically viable protein hydrolysis processes that provide peptides with increased bioactivities (e.g., antihypertensive activity, antioxidant activity) and their use for the development of nutraceutical products, and even as substitutes for synthetic preservatives, continues to emerge [3]. Some studies have suggested using peptides with antihypertensive activity as an alternative to conventional drugs without the adverse effects seen with synthetic formulations [4]. In this case, their use is indicated for the initial treatment of high blood pressure symptoms. Peptides with antioxidant activity also show various potential applications, e.g., in food supplements and therapeutical agents with positive effects on cellular oxidation or as replacements for synthetic antioxidants to inhibit lipid peroxidation in foods during storage [5].

Hydrolyzed proteins can be produced from protein sources of animal (milk, eggs, meat, fish, and insects) [6] or plant origin (soybean, peas, maize, and rice) [7]. In particular, by-product streams from the agri-food industry can provide protein-rich sources for the production of hydrolyzed protein containing peptides with healthy attributes [8,9]. Several studies have demonstrated the production of protein hydrolysates from by-product streams from the dairy (e.g., whey) [10], meat (e.g., skin, viscera, bones) [6,11], and plant processing (e.g., non-edible parts) [12] industries. While some of these studies consider these food protein hydrolysates as sources of antihypertensive peptides, developing processes with maximum efficiency and industrial viability is still challenging [2].

Different methods can be used to produce hydrolysates, i.e., chemical hydrolysis, enzymatic hydrolysis, or microbial fermentation, among which the use of proteases as reaction catalysts are more advantageous compared with chemical processes [9,13]. Protease processes are water-based and do not require chemical solvents, they operate at milder pH and temperature conditions, and they enable a higher process reproducibility and product specificity. Furthermore, it is possible to combine various types of proteases with different specificities, resulting in an increased process efficiency and reaction speed, favoring the production of a hydrolysate with a diverse peptide profile and rich in free amino acids [14,15].

Various factors can influence the efficiency of hydrolysis and the biological properties of hydrolysates, e.g., the type and specificity of the enzyme chosen, the pH, temperature, protein source composition and material properties, and others [9,16]. However, even with the optimization of these parameters, obtaining efficient protein hydrolysis to achieve health and commercial outcomes can be considered a challenge, mainly because of the compact structure of the native state of some proteins [2,17,18]. From this perspective, using high hydrostatic pressure technology (HHP) combined with hydrolysis has been shown to be an option to improve proteolytic efficiencies leading to enhanced bioactivity and positive health attributes [5].

HHP is a technology commercially implemented in several countries in the last decades for non-thermal pasteurization and shelf-life extension purposes. This technology consists of subjecting pre-packaged foods to a hydrostatic pressure ranging during a set holding time, followed by pressure release. High pressure equipment at the laboratory scale exhibits the versatility to operate from 50 to 800 MPa [19,20] for a range of pressure holding times. For current commercial applications, it is commonly employed at 300 or 600 MPa holding for up to 3–5 min, depending on the application [21].

HHP is advantageous due to its ability to cause conformational and structural changes to proteins, particularly by modulating enzymatic processes such as hydrolysis. Proteins have structures maintained by different inter- and intramolecular interactions that can be destabilized using high pressures, resulting in the unfolding of the polypeptide chain [21]. Protein unfolding favors hydrolysis in terms of molecular lysis since more cleavage sites become accessible to enzymes [5,22]. High pressure processing is also known to activate or inactivate hydrolytic enzymes, depending on the HHP and material composition conditions [23]. It has been shown that the characteristics related to the above-mentioned bioactivity of hydrolysates (e.g., antihypertensive activity, antioxidant activity) can also be improved by HHP [5,17,24].

On the other hand, the use of bioactive peptides as ingredients can be limited by the allergenicity of the native protein used to make the hydrolysate. Producing hydrolysates with a greater extent of protein breakdown is an alternative to producing hypoallergenic formulas. In some cases, hydrolytic processing has been shown to achieve the complete destruction of the allergenic epitopes responsible for promoting immunological responses [25]. Furthermore, efforts are underway to understand the effects of HHP in decreasing protein hydrolysate allergenicity [26].

The emerging potential for HHP to enable the production of health-promoting bioactive peptides and to reduce the allergenicity of the substrate makes it worthy of a literature review. The present review examines the effects of HHP on the enzymatic hydrolysis of proteins, with a particular focus on the resulting antihypertensive and antioxidant activities, as well as on the reduction in the allergenicity of protein hydrolysates made from different raw materials.

## 2. Effect of High Hydrostatic Pressure on Proteins

Proteins are macromolecules that have a structural organization that can be divided into primary, secondary, tertiary, and quaternary levels (Figure 1). The balance of these structures is maintained by stabilizing interactions within the protein chain, and any change that may disturb the balance of these interactions, intramolecular and intermolecular, can lead to protein denaturation [27,28].

HHP can promote different levels of protein denaturation, as it acts on protein interactions. Quaternary and tertiary structures are the most affected by the use of pressure, as they are maintained by hydrophobic and electrostatic interactions that are more sensitive to pressure [28,29]. The secondary structure, on the other hand, requires the use of higher pressures for the changes to be effective. Finally, the primary structure is not affected by HHP, as it is formed by covalent bonds [30,31].

For enzymatic hydrolysis, protein unfolding caused by HHP is the most important alteration, being a key parameter to improve the process, as it allows greater access of the enzyme to more specific cleavage points, which were previously protected inside the molecule [32]. However, the parameters used during pressurization, as well as the biochemical characteristics of the proteins, are factors that guide the intensity of denaturation, the rate of refolding after depressurization, and the time for recovery of the initial state of the protein [33].

Furthermore, the formation of molecular aggregates is another important alteration that can be induced by HHP and, contrary to unfolding, can decrease the yield of enzymatic hydrolysis. The disruption of hydrophobic bonds in the core of proteins, as a result of unfolding, causes the exposure of hydrophobic groups and sulfhydryls (-SH), which can remain free to interact and form new hydrophobic interactions and disulfide bonds between proteins [34,35]. Protein–protein interactions, in general, are observed when higher pressures are used and may reduce the number of sites available for enzyme–protein interactions and, as a consequence, may decrease the yield of the hydrolysis process [36,37,38].

## 3. Utilization Strategies of HHP Associated with Enzymatic Hydrolysis

HHP associated with hydrolysis can be carried out in two different ways: (i) as a preliminary protein treatment immediately before hydrolysis or (ii) simultaneously with hydrolysis, as shown in Figure 2A,B, respectively. The use of HHP as a pre-treatment consists of subjecting proteins to high pressure levels for a given pressure holding time and then performing hydrolysis at ambient pressure (0.1 MPa) (Figure 2A). However, using HHP as a pre-treatment can lead to the structural reorganization of proteins after depressurizing the system, leading to the ongoing loss of susceptibility to hydrolysis [33,39,40,41]. More effective hydrolysis has been observed when the enzyme was mixed with the pressurized protein immediately after depressurization [42].

In the simultaneous HHP hydrolysis process, both protein and enzyme are jointly pressurized (Figure 2B). In this case, the enzyme acts by breaking the protein molecules at the moment when pressure promotes the unfolding of the protein structure, which can potentiate hydrolysis since unfolding enables greater enzyme access to the substrate. A more extensive hydrolysis is therefore achieved in a shorter time compared with conventional hydrolysis at ambient pressure (Figure 2C) [32]. Furthermore, the improvement could also be related to enzyme activation since these can be affected (activated or inactivated) by pressure through changes in their conformation and/or selectivity [5,22,43]. However, enzyme stability and sensitivity to pressure and the characteristics of proteins should be considered to ensure that the enzyme is active during the treatment [44].

## 4. Effect of HHP on Proteolysis and on the Bioactivity and Allergenicity of Hydrolysates

### 4.1. Effect of HHP on Protein Hydrolysis

The degree of hydrolysis (DH) of proteins is an important parameter in monitoring the production of protein hydrolysates, as the number of peptide bonds broken in a given proteolysis reaction determines, in quantitative terms, the degree to which a protein source was hydrolyzed [45].

For the evaluated conditions, using HHP as a pre-treatment or simultaneously with hydrolysis is an efficient strategy to increase the DH. Protein unfolding induced by pre-treatment with HHP increases the efficiency of hydrolysis. However, it was observed due to the particularities that each protein can present in relation to the forces that stabilize its structure that the pressure level necessary to influence the efficiency of the process can be different among proteins. Proteins of plant origin, for example, show more expressive DH values when subjected to HHP pre-treatment using pressures close to 300 MPa than at pressures above 400 MPa [36,46,47]. The use of high pressure in plant proteins can favor protein–protein interactions, resulting in a less efficient break of the molecule due to the reduction in the number of available cleavage sites for enzymatic action [47]. On the other hand, some studies have shown that pre-treatment pressures ranging from 400 to 800 MPa favored the production of hydrolysates with a higher DH degree compared with lower pressures, as observed in studies that evaluated the hydrolysis of pre-treated proteins such as β-lactoglobulin [37,48], mushroom stalk proteins [49], and egg proteins [50,51].

In many studies cited in this review, it has been shown that HHP-assisted hydrolysis (AH) is also an efficient strategy to potentiate the catalytic reaction and significantly reduce the processing time, especially compared with the conventional process [32,41,52,53,54]. However, in hydrolysis-assisted treatments, both enzyme and protein are subjected to the same pressure level, and the enzyme’s sensitivity to pressure is a relevant factor that should be considered when evaluating process parameters since the enzyme can undergo changes that result in denaturation and, consequently, the loss of activity. In various studies, the alcalase enzyme, one of the most known in the food production area, increased the DH when subjected to pressures under 200 MPa in different food sources such as lentil proteins [55] and sweet potato protein [53,55,56,57]. On the other hand, the pressure range of 300 to 400 MPa was more efficient in producing hydrolysates from whey proteins, flaxseed protein, and sweet potato protein when the enzymes pepsin and trypsin were the promoting agents of hydrolysis [25,40,52,57].

### 4.2. Effect of HHP on the Antihypertensive Activity of Hydrolysates

High blood pressure is one of the main risk factors for cardiovascular disease, which is one of the most prevalent causes of death in adults [58]. The interest in alternative therapies has grown considerably in recent years, especially due to the adverse reactions that conventional treatments can offer [4]. From this perspective, protein hydrolysates prepared from different food sources have been extensively studied for potentially containing biologically active peptides that can be used to treat high blood pressure [59,60].

Peptides have antihypertensive activity through different mechanisms of action [61], as shown in Figure 3. However, the primary mechanism studied for peptides generated from food sources is inhibiting the angiotensin-converting enzyme (ACE). The renin–angiotensin system is an essential regulator of blood pressure. Renin, secreted by kidney cells, converts angiotensinogen into an inactive peptide, angiotensin I [62]. ACE hydrolyzes this decapeptide into angiotensin II, which has intense vasoconstrictor activity, regulating electrolyte balance and exerting a pro-inflammatory state [62,63] (Figure 3). Furthermore, ACE also acts on the kallikrein–kinin system by promoting the degradation of bradykinin, a peptide known for its vasodilating action [58]. Therefore, inhibition of ACE action is a therapeutic alternative to prevent arterial hypertension, and peptides have been extensively studied regarding their ability to inhibit these systems (Figure 3) [64,65,66].

Although there is evidence showing that HHP is suitable for increasing efficiency in hydrolytic terms, few studies have evaluated the influence of HHP on the antihypertensive activity of hydrolysates, especially compared with the production of conventional hydrolysates, as can be seen in Table 1. Despite this, studies show that the use of HHP associated with hydrolysis increases the yield of peptides with ACE-inhibitory activity and can potentiate these results, as observed in hydrolysates from sweet potato protein [57], soybean [54], common bean [34], lentil [55], peas [67], and eggs [68], as seen in Table 1.

The production of ACE-inhibitor peptides can be influenced by several variables, including the type and source of protein to be hydrolyzed, the enzyme, and its specificity. Furthermore, the parameters employed in the HHP process are also important factors, as can be seen in Table 1. Studies have shown that to obtain hydrolysates with a greater antihypertensive activity, different pressure levels may be required depending on the enzyme used (Table 1). This difference may be related to the fact that each enzyme discussed can respond differently to high pressures, depending on the levels of pressure, temperature, and substrate, showing an increase or decrease in its activity and, consequently, in the production of a functional peptide [23]. 

Moreover, as discussed earlier, HHP disrupts the tertiary and even secondary conformations of proteins, increasing the rate of hydrolysis and favoring the production of greater amounts of bioactive peptides [5,22]. This protein unfolding may also allow the hydrolysis process to occur more quickly [54], and with the use of a lower enzyme:substrate ratio [67], and still present a greater antihypertensive activity compared with the process that does not use HHP. In addition, HHP also contributes to the production of peptides with a lower molecular mass, which is directly correlated with the increase in ACE-inhibitory activity [69].

This has been confirmed by further studies summarized in Table 1 carried out at laboratory scale across a number of substrates, enzymes, and processing conditions. Some processing times seem long and further scaling-up work should consider reducing the processing times, according to the commercial value added to the product. Thus, in general, HHP has been shown to favor the production of hydrolysates with a greater antihypertensive activity in a shorter process time, in some cases with a lower concentration of enzymes. Further cost–benefit analysis is required to demonstrate whether these benefits translate into a scalable and more economical process that will enable commercialization.

### 4.3. Effect of HHP on the Antioxidant Capacity of Hydrolysates

Peptides with antioxidant capacity have received significant attention from the industry and can be incorporated into foods for conservation or improving bioactive properties [54,55,68]. They can act in different ways to inhibit oxidation, mainly through the inactivation of reactive oxygen species, free radical scavenging, chelation of pro-oxidative transition metals, and reduction of hydroperoxides [70].

Among the functionalities of peptides, the antioxidant capacity is the most studied regarding the use of HHP, as shown in Table 2. HHP shows promising applications for the production of hydrolysates with a high antioxidant capacity [53,67]. Different researchers have demonstrated that pre-treatment of proteins or HHP-assisted hydrolysis can increase the antioxidant capacity of hydrolysates (Table 2). In addition, high pressure can also influence the different mechanisms of action of hydrolysates with antioxidant activity. Girgih et al. [38] observed that a pressure of 200 MPa favored the production of hydrolysates with a greater capacity to eliminate superoxide and hydroxyl radicals, while a pressure of 400 MPa was more effective in increasing the ability to sequester the DPPH radical. The different pressure levels used on the protein before hydrolysis, or on the protein and enzymes in assisted hydrolysis, can lead to different structural changes, altering the number of available cleavage sites in the peptide chain, as well as the activity and selectivity of the enzymes, resulting in a change in the profile of generated peptides, and consequently in the mechanism of action, according to the level of pressure used [38,54,55]. This change was observed by Guan et al. [54], who reported that pre-treatment using pressures of 200 and 300 MPa favored the production of peptides not obtained in conventional hydrolysis. This difference can also be observed between the different pressure levels, since at 200 MPa some peptides were identified that were not present in the treatment at 300 MPa.

In the HHP-assisted hydrolysis process, pressure can trigger different effects depending on the enzyme used, as observed by Garcia-Mora [34], who reported that this procedure for bean protein did not increase the donating capacity of hydrogen atoms when the enzyme alcalase was used, whereas the use of savinase in a pressurized system increased the antioxidant capacity of the hydrolysates as pressure increased. On the other hand, pressure potentiated the electron donation capacity of the hydrolysates obtained when using both enzymes, resulting in the highest values for alcalase in assisted hydrolysis at 100 MPa and at 200 MPa for savinase. Similar results were found by Garcia-Mora et al. [55], in whose study the different enzymes used produced hydrolysates with improved antioxidant capacity in different pressure ranges, highlighting the importance of optimizing the time and pressure parameters to which the reaction is subjected, taking into account the enzyme used in the process.

The increased bioactivity of pressurized hydrolysates is evident, although it is still unclear how HHP influences this property. Different studies associate this phenomenon with protein unfolding or enzymatic activation, which potentiates the production of peptides in qualitative and quantitative terms, granting a greater antioxidant capacity through different mechanisms. In addition, the generation of low molecular weight peptides, associated with the release of free amino acids, may also increase the antioxidant capacity of hydrolysates [74,75], since HHP can intensify the presence of free amino acids. Vilela et al. [76] verified a greater quantity and variety of free amino acids in the pressurized hydrolysates, indicating that the pressure also favors a more intense hydrolysis and, consequently, a greater release of amino acids, which may contribute to a greater antioxidant capacity. Several amino acids present antioxidant capacity, but the majority of two reactive amino acids have side chains with a nucleophilic environment (cysteine and methionine) or aromatic side chains (tryptophan, tyrosine, and phenylalanine), which cause hydrogen abstraction [70,77]. Therefore, HHP can intensify antioxidant capacity in the presence of these free amino acids in hydrolysates, as well as influencing their specific position in the peptide sequence, or it can also contribute to their bioactivity.

### 4.4. Effect of HHP on the Allergenicity of Hydrolysates

Food allergy is characterized by a disordered response of the immune system to certain molecules present in food, especially proteins, known as allergens [78]. Many studies have been carried out aiming to reduce the antigenicity of food proteins through different processes that can reduce the presence of allergenic epitopes [79,80]. The epitope is the region of the protein that binds to cell receptors and antibodies. Two types of epitopes can be identified in proteins: linear epitopes, which are specific amino acid sequences located in the primary structure of proteins, and conformational epitopes, which are formed from the tertiary conformation of proteins, in which non-linear amino acid sequences become close, generating the epitope [81].

Conformational epitopes can be disrupted when changes are generated in the tertiary structure or with protein aggregation. However, linear epitopes remain intact from these changes, since the primary structure is not altered [82]. In order to reduce the presence of linear epitopes, it is necessary to use processes that cause changes in the primary protein structure, e.g., enzymatic hydrolysis, which is effective in reducing allergenicity [83]. Nevertheless, food proteins can be resistant to proteolysis, and their fragments can maintain or even increase the sensitization capacity after hydrolysis if the epitope sequence is preserved, despite partial protein hydrolysis [84]. Thus, combining HHP with enzymatic hydrolysis becomes an interesting strategy to reduce or even exempt the allergenic response to various food sources, as shown in Table 3.

Compared with conventional hydrolysis, the association of HHP with enzymatic hydrolysis has shown significant effects in reducing the allergenicity of whey proteins [87,88], in addition to studies showing that, even without hydrolysis, HHP alone can reduce the allergenic response [29].

The strategy of using HHP for protein hydrolysis can affect the immunochemical characteristics of the hydrolysates, as reported by Chicón et al. [86], in whose study the pre-treatment of β-lactoglobulin by HHP showed no difference in immunoreactivity compared with the conventional hydrolysate. However, when the hydrolysis was simultaneously conducted with the use of HHP at the pressure of 400 MPa for 20 min, the hydrolysates did not show significant IgE immune responses [39]. Additionally, in the case of using HHP as a previous treatment of proteins, pressures above 600 MPa followed by hydrolysis can also reduce the in vitro reactivity with IgE of allergic individuals [89]. Landim et al. [85] verified that pre-treatment of whey proteins at 400 MPa was important to decrease the hydrolysis time necessary to reduce 100% of the immunoreactivity of the hydrolysates produced by the Novo Pro-D enzyme, also favoring a greater reduction in the immunoreactivity of the hydrolysates obtained by the ficin enzyme compared with the hydrolysate obtained in the conventional process.

In addition, the time of hydrolysis is another important parameter. In general, the use of HHP may favor the hydrolysis of the native protein in the first minutes of the reaction. However, it may not be sufficient to reduce the immunoreactivity of hydrolysates, since the protein fragments generated at the beginning of the reaction may still contain the immunoreactive epitopes intact [90]. López-Expósito [90] reported that the immunoreactivity of ovalbumin proteins hydrolyzed under pressure decreased with increasing treatment times but still maintained a small perceptible residual binding to IgE in egg-allergic patients. Similar results were found by Chicón et al. [39,86], in whose studies whey proteins hydrolyzed at 400 MPa showed progressive antigenicity reduction with treatment time, especially in relation to non-pressurized protein hydrolyzed at ambient pressure. However, reactions against IgE were still noticeable.

HHP changes the conformation of proteins that alter conformational epitopes, potentiating enzymatic hydrolysis and, consequently, reducing the presence of linear epitopes [26,29,84]. However, even after carrying out the protein hydrolysis process, it is still necessary to evaluate the immunoreactivity to IgE of the peptides formed in order to guarantee the reduction or exemption of an allergenic response. Studies in vivo conducted by López-Expósito [91] in sensitized mice proved that β-lactoglobulin protein hydrolysates under high hydrostatic pressure (400 MPa) are immunologically inert. In a study conducted by Lozano-Ojalvo [25], whey protein hydrolysates treated at 400 MPa showed allergenic epitopes of β-lactoglobulin and α-lactalbumin. However, changes in the immune response were not perceptible in the tests performed in vivo, e.g., changes in body temperature, anaphylaxis, and signs or release of markers in mice sensitized with the allergen. The study showed that HHP can produce hypoallergenic hydrolysate in a short time, i.e., which does not promote reactions mediated by IgE and IgG.

Finally, protein unfolding induced by HHP treatment facilitates enzymatic hydrolysis, resulting in shorter peptide fragments, with between 7 and 10 amino acid residues, which may not be recognized as epitopes, leading to reduced allergenicity [48,92]. Therefore, it is possible to obtain peptides with a lower allergenic response using HHP combined with enzymatic hydrolysis, since the peptide pattern and the immunoreactivity of the hydrolysates obtained under HHP can be altered by selecting the enzyme, pressure, and time of hydrolysis [25,91].

## 5. Conclusions

High hydrostatic pressure, when combined with enzymatic hydrolysis, is an emerging process that shows promising results in the production of protein hydrolysates. The use of HHP, either as a pre-treatment or simultaneously with hydrolysis, is an alternative to improve the process efficiency. In general, the combination of HHP and enzymatic hydrolysis potentiates catalysis in different protein sources, in a significantly shorter process time and allowing the use of a smaller amount of enzyme in relation to the substrate compared with conventional hydrolysis carried out at ambient pressure. Furthermore, the protein hydrolysates obtained show improved properties in terms of antihypertensive activity, mainly regarding the in vitro ACE-inhibitory activity and greater in vitro antioxidant capacity. HHP is also an important technology in the production of hypoallergenic hydrolysates. However, in order to obtain optimal results, it is important to individually evaluate the processing parameters in relation to the HHP technology, e.g., the way in which it is used (pre-treatment or simultaneously with hydrolysis), the selection of the enzyme, and the process parameters (pressure level and processing time), since all these factors can significantly impact the allergenic response. In addition, most of the research that has evaluated the effect of HHP on the health-promoting properties of hydrolysates has used in vitro assays. Thus, it becomes important to use clinical trials to provide evidence that the claimed properties are actually enhanced and have in vivo effects, making protein hydrolysates ingredients with an even greater market potential for the industry that seeks to meet a niche of consumers who are increasingly concerned about the relationship between health and food.

## Figures and Tables

**Figure 1 foods-12-00630-f001:**
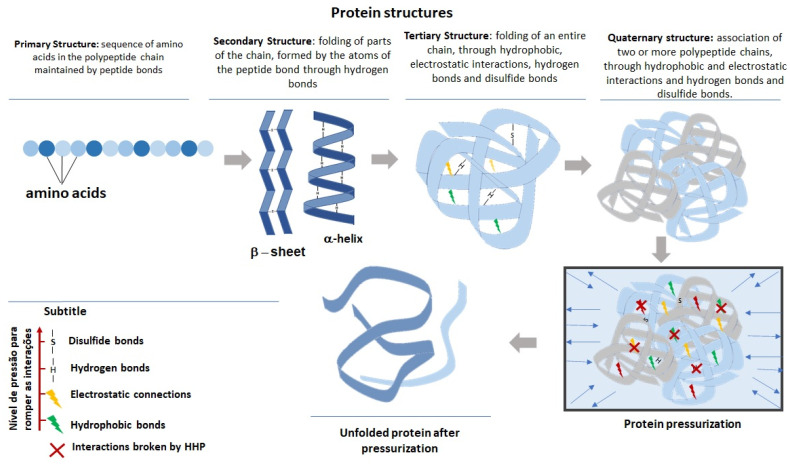
Different structural levels of proteins and the effect of high hydrostatic pressure on the structure and conformation of proteins.

**Figure 2 foods-12-00630-f002:**
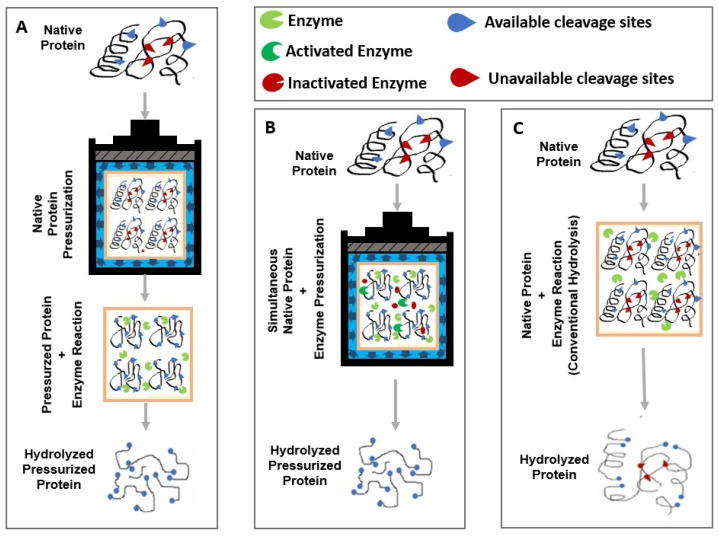
Strategies of HHP application to improve enzymatic hydrolysis. (**A**) Protein pre-treatment followed by enzymatic hydrolysis (pre-treatment—PT); (**B**) HHP treatment simultaneously with enzymatic hydrolysis (assisted hydrolysis—AH); and (**C**) conventional hydrolysis at an ambient temperature.

**Figure 3 foods-12-00630-f003:**
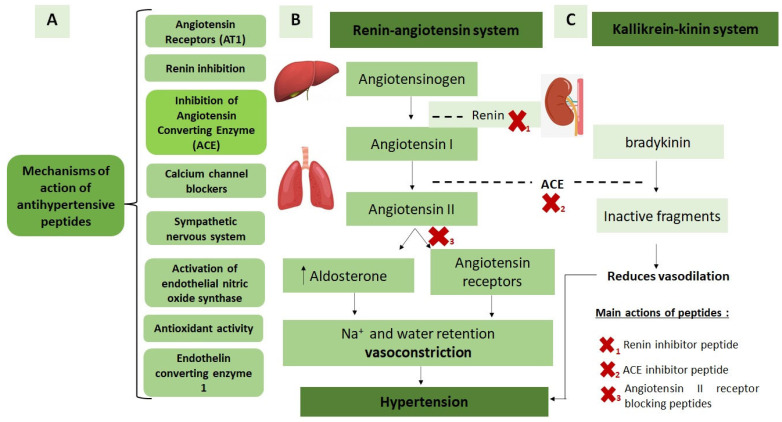
Mechanisms of action of antihypertensive peptides (**A**); regulation of blood pressure by the renin–angiotensin (**B**) and kallikrein–kinin (**C**) systems.

**Table 1 foods-12-00630-t001:** Influence of HHP on the production of hydrolysates with antihypertensive activity.

Source	HHP Parameter (Pressure/Time)	Enzyme	Main Results	Reference
Sweet potato	100, 200, and 300 MPa/60 min (AH)	Papain, Pepsin, and Alcalase	Pressure increased the ACE-inhibitory activity in all treatments. However, for the papain and alcalase hydrolysates, the maximum inhibitory activity was obtained with pressures lower than 200 MPa and below 300 MPa for pepsin.	[57]
Soybean	80, 100, 120, 200, and 300 MPa/1, 2, 3, 4, and 5 h (AH)	Colorase PP^®^	The hydrolysates obtained under high pressure showed a higher ACE-inhibitory activity than conventional hydrolysis. The treatment at 200 MPa for 5 h resulted in the greatest inhibition.	[54]
Common bean	100 and 200 MPa/15 min (AH)	Alcalase and Savinase	HHP-assisted hydrolysis increased the ACE-inhibitory activity at 100 and 200 MPa for alcalase and at 200 MPa for savinase, in addition to reducing the hydrolysis time.	[34]
Lentil	100, 200, 300, and 400 and 500 MPa/15 min (AH)	Alcalase, Savinase, Promatex, and Colorase PP^®^	The treatment had no effect on the hydrolysate obtained from alcalase. The pressure of 200 MPa for promatex and 300 MPa for colorase and savinase resulted in hydrolysates with a higher ACE-inhibitory activity.	[55]
Peas	200, 400, and 600 MPa/5 min (PT)	Alcalase	Pre-treatment at 600 MPa favored an increase in peptides with ACE-inhibitory capacity, even when using low enzyme concentrations.	[67]
Eggs	100–400 MPa/5, 10 and 20 min (AH) 100–400 MPa/20 min (PT)	Pepsin, Chymotrypsin, and Trypsin	High pressure hydrolysis accelerated the process and increased the release of peptides identified with ACE-inhibitory activity.	[68]

PT—pre-treatment; AH—assisted hydrolysis; ACE—angiotensin-converting enzyme.

**Table 2 foods-12-00630-t002:** Influence of HHP on the production of hydrolysates with antioxidant capacity.

Source	HHP Parameter (Pressure/Time)	Assay	Enzyme	Main Results	Reference
Whey	100, 250, 400 MPa/5, 20, and 35 min (AH and PT)	ORAC and ABTS	Pepsin	PT resulted in hydrolysates with a higher antioxidant capacity than conventional treatment.	[32]
Lentil	300, 400, and 600 MPa/15 min (PT)	DPPH	Alcalase	PT at 300 MPa/15 min increased the hydrolysate’s ability to reduce the DPPH radical.	[46]
Common bean	300, 400, and 600 MPa/15 min (PT)	DPPH	Alcalase	PT at 300 MPa produced hydrolysates with a greater antioxidant capacity.	[36]
Soybean	80, 100, 120, 200, and 300 MPa/1, 2, 3, 4, and 5 h	ABTS Reducing power	Colorase PP^®^	The treatment with 200 MPa/5 h resulted in a hydrolysate capable of reducing the ABTS radical by 62%.	[54]
Flaxseed	100 and 300 MPa/5 and 10 min (AH)	ORAC	Trypsin	Pressure and time were important factors in increasing bioactivity, with treatment at 300 MPa/10 min increasing the antioxidant capacity by 20%.	[52]
Phosvitin	50 and 100 MPa/6, 12, and 24 h (AH)	DPPH, FRAP, SRSA, and MCA	Alcalase, trypsin	HHP improved the ability to reduce the DPPH radical, the superoxide radical scavenging, and the iron reduction capacity of the alcalase hydrolysate. The iron chelation capacity was improved for alcalase and trypsin.	[71]
β-lactoglobulin	100, 200, 300, and 400 and 500 MPa/15 min (AH)	ORAC	Alcalase, Savinase, Promatex, and colorase	HHP improved the antioxidant capacity of the hydrolysates.	[72]
Flaxseed	600 MPa/5, 10, and 20 min (PT)	ORAC	Trypsin and trypsin-pronase	PT at 600 MPa/20 min resulted in the greatest increase in the antioxidant activity of the hydrolysates.	[47]
Sweet potato	100, 200, and 300 MPa/ 30 and 60 min (AH)	ORAC	Alcalase	The hydrolysate obtained in the HHP-assisted hydrolysis using 300 MPa/20 min resulted in the highest antioxidant capacity.	[7]
Common bean	100 and 200 MPa/15 min (AH)	ORAC, FRAP, ABTS	Alcalase and Savinase	The highest ORAC and ABTS values of the hydrolysates were observed in the treatments at 200 MPa for savinase and 100 MPa for alcalase. HHP had no influence on the ability of the hydrolysates to reduce iron for both enzymes.	[34]
Lentil	100, 200, 300, 400, and 500 MPa/15 min (AH)	ORAC	Alcalase, Savinase, Promatex. Colorase	Treatment at 100 MPa produced hydrolysates with a higher antioxidant capacity from alcalase, and 300 MPa exhibited greater effects for savinase and colorase.	[55]
Peas	200, 400, 600 MPa/5 min (PT)	ORAC, DPPH, FRAP, MCA, SRSA	Alcalase	Improved DPPH scavenging capacity at 400 MPa. Improved ORAC activity at 400 and 600 MPa.	[38]
Chickpea	100, 200, 300, 400, 500, and 600 MPa (PT); 100, 200, 300 MPa (AH)	SRSA, FRAP	Alcalase	PT 300 and 400 MPa and assisted hydrolysis at 200 MPa/30 min were more effective in increasing the antioxidant capacity of hydrolysates.	[73]

PT—pre-treatment; AH—assisted hydrolysis; MCA—metal chelating activity; SRSA—superoxide radical scavenging activity; ORAC—oxygen radical absorbance capacity; FRAP—ferric reducing antioxidant power; DPPH—DPPH radical method; ABTS—ABTS radical method.

**Table 3 foods-12-00630-t003:** Influence of HHP on the production of hydrolysates with reduced allergenicity.

Source	HHP Parameter (Pressure/Time)	Enzyme	Main Results	Reference
Whey	100, 250, 400 MPa/5, 20, and 35 min (PT)	Novo Pro-D and Ficin	PT contributes to reducing the antigenicity of the hydrolysates obtained by ficin and reduced the hydrolysis time from 60 min to 15 min necessary to achieve a complete reduction in immunoreactivity.	[85]
Whey	400 MPa/30 min (AH)	Pepsin	Maximum hydrolysis with peptides ranging from 10 to 3 kDa (50%), with a reduction in intact allergens and no induction of clinical signs in sensitized mice.	[25]
β-lactoglobulin	100, 200, 300, and 400 MPa/120 min (PT and AH)	Pepsin	PT did not influence the immunoreactivity of hydrolysates. Assisted hydrolysis progressively reduced antigenicity with increasing incubation times and pressures.	[86]
Whey	200 and 400 MPa/10, 30, and 60 min (AH)	Pepsin and Chymotrypsin	HHP favored a reduction in b-LG immunoreactivity with the use of the pepsin enzyme, which was progressive with the incubation time and increasing pressure (400 MPa/30 min).	[39]
Whey	100 and 200 MPa/min (PT and AH)	Alcalase, Neutrase, Colorase 7089, and Colocarase PN-L	The treatment influenced the reduction in antigenicity only in the hydrolysate obtained from the enzyme Colorase PN-L at 300 MPa.	[87]
Soybean	100,200, and 300 MPa/15 min (AH)	Alcalase, Neutrase, and Colorase	The treatment using the pressure of 300 MPa contributed to reducing the immunoreactivity of the hydrolysates obtained from Colorase.	[88]
β-lactoglobulin	600 MPa/10 min (PT)	Trypsin and Chymotrypsin	Decreased immunoreactivity after combination treatment, which was greater when chymotrypsin was used.	[89]

PT—pre-treatment; AH—assisted hydrolysis.

## Data Availability

Not applicable.

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
