# Peer review of "Potential Application of High Hydrostatic Pressure on the Production of Hydrolyzed Proteins with Antioxidant and Antihypertensive Properties and Low Allergenicity: A Review"

_foods, 2023, doi:10.3390/foods12030630_

Round 1

Reviewer 1 Report

The MS is well-written and is a good contribution to the field.

The MS provides an update on the use of HP to generate peptides and the technology's impact on the biological activities of the peptides/ hydrolysates. The topic is not extensively reviewed in the literature and thus it has an element of novelty.

Compared with other published material, I am aware of two reviews on the topic that have been published in Trends in Food Sci that are at least 2 years old. None of those covers the impact on allergenicit.

Sections 3.2 and 3.3 are not relevant to the topic and have been reported elsewhere. Unless the authors make a direct relation between these sections and  allergenicity, they should be removed.

The MS is clear and easy to read and contains very useful artwork.

Author Response

We appreciate your comment. However, we would like to keep topics 3.2 and 3.3 of our review as they are an important part of the review purpose, which was to cover the biological and allergenic responses of the hydrolysates using HHP. The choice to address the antihypertensive activity and antioxidant capacity can be justified as they are the main known biological properties studied in food matrices.

Reviewer 2 Report

This paper reviews the effects of high hydrostatic pressure technique on the biological properties and allergenicity of proteins after hydrolysis. Although the booming interest on HHP in the Food industry could justify their attention in this research work, I am afraid that the existence of several flaws in this manuscript makes it unsuitable. Some major aspects need to be addressed.

1. In Abstract, the description should focus on the technology, not the protein.

2. Comparative representation of protein hydrolysis by various methods is not adequate.

3. The title of this article is "Use of high hydrostatic pressure processing to produce protein hydrolysates with biological properties and reduced allergenicity: a review", mentions "biological properties", but in the description below only the anti-hypertensive and antioxidant capacity of the protein after hydrolysis is addressed. Is it a bit of a generalization?

4. Why the focus on anti-hypertensive substances?

Author Response

In Abstract, the description should focus on the technology, not the protein.

A: Thank you very much for your comment. The abstract was rewritten and more emphasis on the HHP process was given by reorganizing and adding more information to the text on page 1 and lines 16 – 24.

Comparative representation of protein hydrolysis by various methods is not adequate.

A: The authors did not clearly understand the reviewer comment. Alternatively, we are proposing the use of another Figure 1 replaced on the revised manuscript, as an option that may clarify the content of the text.

The title of this article is "Use of high hydrostatic pressure processing to produce protein hydrolysates with biological properties and reduced allergenicity: a review", mentions "biological properties", but in the description below only the anti-hypertensive and antioxidant capacity of the protein after hydrolysis is addressed. Is it a bit of a generalization?

A: In short, most studies that evaluated the use of HHP in the enzymatic hydrolysis of proteins focused on trying to understand its effects only on antioxidant and antihypertensive activities and, for this reason, the review also had this purpose. However, we understand that this may raise a generalization idea at first, so we have changed the title to specify the two biological activities addressed on the manuscript.

Why the focus on anti-hypertensive substances?

A: The anti-hypertensive activity of peptides has been widely studied in recent years, mainly due to the possibility of offering an alternative therapy to conventional blood hypertension treatments. Some peptides with this functionality are already commercially available in different countries. In this sense, the optimization of hydrolysis to obtain peptides with more anti-hypertensive activity assumes an important role in this area, leading to efforts to search more efficient processes, by evaluating different protein matrices, enzymes, and technologies.

Reviewer 3 Report

The manuscript treat about very interesting topic, few studied. It is possible that more discussion would be required or more deep analisys on some part inside of text, because on some times the authors just describe the results from literature without   analisys about that. On the other hand, there are many sentences without references in which they made importants asseverations.

Author Response

The manuscript treat about very interesting topic, few studied. It is possible that more discussion would be required or more deep analisys on some part inside of text, because on some times the authors just describe the results from literature without   analisys about that. On the other hand, there are many sentences without references in which they made importants asseverations.

A: We thank you very much for your considerations and notes included on the pdf of the manuscript. We have taken all under consideration and the changes are market in red all through the revised version of the text.

An extensive revision of the references were made and added to the pointed paragraphs and sentences.

Figure 1 was replaced.

On page 4 and lines 164-166, references were added, and a sentence: "cited in this study".

Sentences on page 6 and lines 243-251 have been removed.

Paragraph on page 9, lines 310-314 have been changed to improve the understanding.

On page 11 and lines 322-326, the paragraph that was repeated has been removed.

Round 2

Reviewer 2 Report

The revised manuscript has been modified accordingly. All the questions that reviewer mentioned has been well answered.

Author Response

We are grateful to the reviewer comments and consideration.

Reviewer 3 Report

All observations that I made in first version were taking in consideration.